# A Pyrazolate Osmium(VI) Nitride Exhibits Anticancer Activity through Modulating Protein Homeostasis in HepG2 Cells

**DOI:** 10.3390/ijms232112779

**Published:** 2022-10-24

**Authors:** Chengyang Huang, Wanqiong Huang, Pengchao Ji, Fuling Song, Tao Liu, Meiyang Li, Hongzhi Guo, Yongliang Huang, Cuicui Yu, Chuanxian Wang, Wenxiu Ni

**Affiliations:** 1Department of Physiology, Shantou University Medical College, Shantou 515041, China; 2Department of Biological Chemistry, Eli and Edythe Broad Center for Regenerative Medicine and Stem Cell Research, David Geffen School of Medicine, University of California at Los Angeles, Los Angeles, CA 90095, USA; 3Department of Medicinal Chemistry, Shantou University Medical College, Shantou 515041, China

**Keywords:** osmium complex, pyrazole derivatives, anticancer, protein homeostasis, apoptosis

## Abstract

Interest in the third-row transition metal osmium and its compounds as potential anticancer agents has grown in recent years. Here, we synthesized the osmium(VI) nitrido complex **Na[Os^VI^(N)(tpm)_2_]** (tpm = [5-(Thien-2-yl)-1H-pyrazol-3-yl]methanol), which exhibited a greater inhibitory effect on the cell viabilities of the cervical, ovarian, and breast cancer cell lines compared with cisplatin. Proteomics analysis revealed that **Na[Os^VI^(N)(tpm)_2_]** modulates the expression of protein-transportation-associated, DNA-metabolism-associated, and oxidative-stress-associated proteins in HepG2 cells. Perturbation of protein expression activity by the complex in cancer cells affects the functions of the mitochondria, resulting in high levels of cellular oxidative stress and low rates of cell survival. Moreover, it caused G2/M phase cell cycle arrest and caspase-mediated apoptosis of HepG2 cells. This study reveals a new high-valent osmium complex as an anticancer agent candidate modulating protein homeostasis.

## 1. Introduction

Cancer is a leading cause of death worldwide [1] and chemotherapy remains the outstanding effective strategy for prolonging patient survival. The clinical success of platinum-based anticancer drugs such as cisplatin, carboplatin, and oxaliplatin has stimulated extensive investigations into new metallodrugs with improved pharmacological properties, which may reduce side effects, such as kidney toxicity and nausea, and overcome drug resistance [2,3,4]. Recently, other metal-based compounds with potential anti-cancer properties have been reported [5,6]. Ruthenium compounds are a potential alternative to platinum-based drugs owing to their chemical and pharmacological properties. A number of ruthenium-based complexes have been reported to have promising anticancer activity, two of which are still in clinical trials [7,8]. Osmium offers several features that are distinct from ruthenium, including a preference for higher stable oxidation states, stronger p back-donation from lower oxidation states, stronger spin-orbit coupling, and slower ligand exchange kinetics. In addition, osmium is relatively inert and stable under physiological conditions, making it a promising anticancer agent candidate. Sadler et al. reported a library of half-sandwich “pianostool” osmium(II) arene complexes that displayed increased anticancer activity [9]. Our previous data showed that osmium(VI) nitrido compounds with tridentate Schiff bases [10,11] and monodentate azole heterocycle ligands had anticancer properties in vitro and in vivo by targeting DNA strands [12]. Recently, we reported a new osmium(VI) nitrido complex bearing a nonplanar tetradentate ligand with potent anticancer activity. It causes mitochondrial damage and induces liver cancer cell bimodal death via oncosis and apoptosis [5]. Lippard et al. found that the cellular response evoked by anti-proliferating osmium(VI) nitrido compounds with bidentate lipophilic N,N-chelating ligands could be tuned by subtle ligand modifications [13]. In addition, osmium(VI) nitrido complexes more effectively inhibited the growth of breast cancer stem cells compared with platinum-based anticancer drugs, suggesting that osmium(VI) nitrido complexes might decrease tumor survival, proliferation, metastasis, and recurrence [14,15,16,17,18].

Herein, we report and characterize a new osmium(VI) nitrido complex **Na[Os^VI^(N)(tpm)_2_]** (tpm = [5-(thien-2-yl)-1H-pyrazol-3-yl]methanol), which has two bidentate pyrazolate ligands. It undergoes different anticancer mechanism from cisplatin and our previous nitrido-osmium complexes. We investigated its anticancer activity in seven human cancer cell lines, including one cancer stem cell line and one cisplatin-resistant cell line. Proteomic analysis suggested that HepG2 cells’ treatment with **Na[Os^VI^(N)(tpm)_2_]** regulated protein homeostasis decreased the proteomic pathway of DNA repair and increased the pathways of DNA damage and DNA replication pressure, resulting in oxidative stress evidenced by an increase in reactive oxygen species (ROS) and a reduction in mitochondrial membrane potential (MMP). The complex arrested the cell cycle in the G2/M phase and activated the caspase-mediated apoptosis pathway of cancer cells. Furthermore, the anticancer activity of **Na[Os^VI^(N)(tpm)_2_]** was examined in an animal model, showing that the tumour significantly decreased in vivo.

## 2. Results and Discussion

### 2.1. Synthesis and Characterization of the **Na[Os^VI^(N)(tpm)_2_]** Complex

Pyrazole derivatives are important pharmacophores in medicine that display diverse pharmacological activities, such as antitumor, antioxidant, antibacterial, and anti-inflammatory activities. Therefore, a large number of transition metal complexes have been synthesized and applied in medicine [19,20]. Herein, we synthesized a new anionic osmium(VI) nitrido complex bearing a pyrazolate ligand. The reaction of (^n^Bu_4_N)[Os^VI^(N)Cl_4_] [21] with two equivalents of [5-(Thien-2-yl)-1H-pyrazol-3-yl]methanol (tpm) in methanol took place in the presence of NaOH. The product **Na[Os^VI^(N)(tpm)_2_]** was characterized by elemental analysis, ^1^H NMR spectroscopy, infrared spectroscopy, electrospray ionization mass spectrometry and ICP-MS (Appendix A, Appendix A). Additionally, the X-ray crystal structure of **Na[Os^VI^(N)(tpm)_2_]** was determined (Figure 1 and Appendix A). It showed that the osmium centre is coordinated by two deprotonated tpm ligands and one nitrido ligand. The Os≡N bond distance is 1.649(9) Å and the IR stretch for Os≡N occurs as a medium band at 1107 cm^–1^, which are comparable to other osmium nitrido complexes [22]. The stability of the complex was observed by UV/Visible spectroscopy (Appendix A). It is stable in DMSO for 72 h; however, the absorption peak blue shifts and declines slightly in PBS (phosphate-buffered saline, pH 7.0) containing 3% DMSO. It should be noted that the ligand pyrazolate N has been deprotonated owing to its reaction with osmium salt in the presence of NaOH. When **Na[Os^VI^(N)(tpm)_2_]** is dissolved in PBS, protonation is preferred at the ligand N, changing the absorption. On the other hand, the 30 μM complex is stable in the presence of the high concentration biological reductant glutathione (GSH), showing the similar blue shift in UV/Visible spectra (Appendix A).

### 2.2. Anticancer Ability of the Na[Os^VI^(N)(tpm)_2_]

The anticancer ability of **Na[Os^VI^(N)(tpm)_2_]** was investigated by screening its antiproliferation in seven cancer cell lines, including HeLa (cervical), HepG2 (liver), A549 (lung), A2780 (ovarian), and MDA-MB-231 (breast) cells, as well as a liver cancer stem cell line (HepG2-stem) and a cisplatin-resistant cell line (A549CIS) [23,24]. As shown in Table 1, the complex exhibited significant anti-cancer activity with IC_50_ values in the range of 5.6–15.3 µM. The antiproliferation of clinically used cisplatin after 48 h of treatment was also examined for comparison [11]. We found that the complex robustly inhibited the growth of both the HepG2 and HepG2-stem cell lines, although it had a greater inhibitory effect on the cell viabilities of the cervical, ovarian, and breast cancer cell lines compared with cisplatin. We have also evaluated the antiproliferation of complex towards human normal liver cell line (LO-2). It was found that the compound is less cytotoxic to normal cells, with an IC_50_ value of 28.5 μM, than cancer cells. Notably, the compound can also reduce the proliferation of cisplatin-resistant cells. This compound is nearly as effective in cisplatin-resistant cells, with a resistance factor (RF) of 1.3. The RF determined for cisplatin was 12.2, which is 9.4-fold greater than that of **Na[Os^VI^(N)(tpm)_2_]**.

### 2.3. Proteomics Analysis

To gain insight into the molecular effects of **Na[Os^VI^(N)(tpm)_2_]** on HepG2 cells, we performed proteomic analysis of cells after 8 h and 24 h of treatment with **Na[Os^VI^(N)(tpm)_2_]** by mass spectrometry. Gene Ontology (GO) analysis revealed that the expression of proteins in the cells after 8 h and 24 h of treatment had distinct proteomic features (Figure 2A). After 8 h of treatment with **Na[Os^VI^(N)(tpm)_2_]**, the expression levels of proteins involved in protein transportation and localization decreased by at least 1.5-fold, while proteins involved in cytoskeletal organization and morphogenesis of the epithelium increased by at least 1.5-fold, indicating that **Na[Os^VI^(N)(tpm)_2_]** altered the dynamics of the cells and affected cell infiltration. Interestingly, 24 h of treatment with **Na[Os^VI^(N)(tpm)_2_]** decreased the expression of proteins involved in DNA metabolism, the cell cycle, nucleic acid metabolism, and DNA damage repair, while increasing protein expression for the oxoacid, amide, and antibiotic metabolic process and the response to unfolded proteins (Figure 2A). The top categories from the GO analysis of up- and downregulated proteins after 24 h of treatment with **Na[Os^VI^(N)(tpm)_2_]** were oxoacid and DNA metabolic processes, respectively. All of the proteins in these two categories are shown in the heat map (Figure 2B). Proteins involved in oxoacid metabolism play key roles in cancer treatment [25,26]. The DNA metabolic process contains a large number of proteins associated with DNA replication and DNA damage, which may be the molecular mechanisms behind the observed increased cell apoptosis after drug treatment [27].

In the case of the DNA metabolic process, the abundances of 27 proteins were significantly altered. These proteins play a role in DNA replication and repair (Figure 2B). Reduced expression of these genes may lead to the accumulation of DNA damage to cells and reduce cell proliferation, which in turn promotes cell apoptosis [27]. INTS3 can combine with hSSB1 to form a complex to regulate the stability of p53 [28]. The INTS3 gene is highly expressed in many hepatocellular carcinoma cell lines. SMARCA1 is associated with chromatin remodelling. Mutations of this gene in HCC tumours may significantly contribute to the occurrence of hepatocyte tumours [29]. TOP2A (topoisomerase II α) is overexpressed in HCC cell lines. TOP2A is a cellular topoisomerase that determines the tumour cell response to chemotherapeutics and can also be used as a biomarker of drug resistance in cancer [30,31,32].

Moreover, DDB2 expression greatly decreases after **Na[Os^VI^(N)(tpm)_2_]** treatment. DDB2 interacts with DDB1 to form a UV-DDB complex that senses UV-induced DNA damage and initiates DNA repair through nucleotide resection (the NER pathway) and repair [33]. Interestingly, DDB2 can inhibit the proliferation and migration of cancer cells mediated through PAQR3 [34]. In addition, DDB2 directly interacts with LRH-1 and facilitates LRH-1 protein ubiquitination and degradation, which are involved in sugar and lipid metabolism and protein ubiquitination [35]. The decreased expression of DDB2 after **Na[Os^VI^(N)(tpm)_2_]** treatment suggests that these DDB2 pathways may be impaired, resulting in cellular damage.

TYMS (thymidine synthase), the only enzyme involved in folic acid metabolism [36], is essential for regulating normal DNA replication and is an important target for some chemotherapy drugs. Thymidylate synthase is induced by the transcription factor LSF/TFCP2, which plays an important role in the proliferation and drug resistance of hepatocellular carcinoma [37]. TYMS can be inhibited by drugs that inhibit DNA synthesis and replication [38]. Thymidine synthase, four methylene folate reductases, two hydrogen pyrimidine dehydrogenases, and thymidine phosphatase are key enzymes that determine the sensitivity or resistance to drugs [39].

In the case of the oxoacid metabolic process, the expression of GCLM increased greatly (Figure 2B). The glutamate-cysteine ligase regulatory subunit is encoded by the GCLM gene and is the first rate-limiting enzyme in glutathione synthesis. The enzyme consists of two subunits, a heavy catalytic subunit (glutamate-cysteine ligase regulatory subunit (GCLM)) and a light regulatory subunit (glutamate-cysteine ligase catalytic subunit (GCLC)). In response to various oxidative stresses, GCLM can be regulated through the electrophile response element (EPRE) [40]. EPRE has been found to be an indicator of oxidative stress [41]. Our data suggest that **Na[Os^VI^(N)(tpm)_2_]** regulates the cellular redox state to affect the survival of cancer cells.

Another protein that is highly upregulated after **Na[Os^VI^(N)(tpm)_2_]** treatment is KYNU. The kynurenine pathway is the major metabolic pathway of the amino acid tryptophan. Kynureninase or L-kynurenine hydrolase (KYNU) is an important enzyme in this pathway. It is a PLP-dependent enzyme that catalyses the cleavage of kynurenine (Kyn) into anthranilic acid (Ant) [42]. In some cancer transcription analyses, KYNU often has abnormal expression and, in non-targeted metabolomics analysis, KYNU in the kynurenine pathway is regarded as a potential chemotherapy target [43]. KYNU was downregulated in highly invasive cell lines, which was associated with tumour inhibition in osteosarcoma. In addition, KYNU significantly decreased in most cancer cells and appeared to be associated with the infiltration of cancer cells [44].

Overall, these data consistently suggest that treatment of cancer cells with **Na[Os^VI^(N)(tpm)_2_]** decreases the DNA repair capacity while increasing DNA damage accumulation and DNA replication pressure, resulting in cellular oxidative stress and cell apoptosis.

### 2.4. Effects of **Na[Os^VI^(N)(tpm)_2_]** on Cell Cycle and Apoptosis

To determine whether **Na[Os^VI^(N)(tpm)_2_]** triggers the mechanisms of cell cycle arrest and apoptosis, we examined the counteraction of **Na[Os^VI^(N)(tpm)_2_]** on cell cycle progression in HepG2 cancer cells by flow cytometry analysis (Figure 3A). The cells were treated with the complex at 3.25, 6.5, and 13 μM. Compared with the control, treatment with 3.25 μM for 12 h caused a slight increase in the percentage of cells in the G1 phase from 63.65% to 67.67%. Treatment with **Na[Os^VI^(N)(tpm)_2_]** at 13 μM led to mainly G2/M phase arrest, where the percentages of cells increased from 8.47% to 15.72%. We further examined the expression levels of key proteins that promote the G2/M phase transition. The level of cyclin B1, which is responsible for the cell cycle transition from the G2 phase to the M phase, binds to the activated form of CDC2 to form a complex to ensure that the cells enter the M phase properly [45,46].

Western blot analysis showed that the expression levels of cyclin B1 and p-CDC2 remarkably increased after treatment with the **Na[Os^VI^(N)(tpm)_2_]** (4, 8, and 12 μM). Our data suggest that the **Na[Os^VI^(N)(tpm)_2_]** compound may cause G2/M phase arrest by inducing the formation of the cyclin B1/p-CDC2 complex to perturb the protein dynamics of the cell cycle (Figure 3B).

In order to investigate whether **Na[Os^VI^(N)(tpm)_2_]** triggers cell apoptosis, HepG2 cancer cells and HepG2-stem cancer stem cells were treated with **Na[Os^VI^(N)(tpm)_2_]** for 24 h and subjected to flow cytometry analysis. Apoptosis is a programmed cell death process and many metal-based anticancer drugs have been reported to be anti-proliferative by inducing apoptosis [47,48,49]. As shown in Figure 4A, in an **Na[Os^VI^(N)(tpm)_2_]** concentration-dependent manner, the apoptotic (early and late apoptotic cells) proportion of HepG2 cancer cells increased from 5.21% to 38.35%, where late apoptosis was more significant. Compared with the control treated with DMSO, apoptosis of HepG2-stem cells treated with the **Na[Os^VI^(N)(tpm)_2_]** significantly increased from 5.02% to 64.9%, which was attributed equally to both early and late apoptosis. These results suggested that cell death after treatment with **Na[Os^VI^(N)(tpm)_2_]** was mainly induced through apoptosis. Previous studies have shown that caspases are a family of cysteinyl aspartate-specific proteases comprising 12 human members, most of which play key roles in programmed cell death. Caspases-8/9 are the initiator caspases of extrinsic apoptosis, while caspase-3 is an executioner of apoptosis [50]. In addition, cleavage of poly-(ADP-ribose)-polymerase-1 (PARP-1) is one of the imperative indicators for caspase-mediated apoptosis [51]. To investigate whether **Na[Os^VI^(N)(tpm)_2_]**-induced apoptosis is mediated by caspases, we examined the protein levels of caspases and cleaved-PARP-1 in **Na[Os^VI^(N)(tpm)_2_]**-treated HepG2 cells by Western blotting. As shown in Figure 4B, **Na[Os^VI^(N)(tpm)_2_]** displayed the sequential activation of caspases 9 and 3, which indicated that **Na[Os^VI^(N)(tpm)_2_]** can induce cell apoptosis [52]. From the perspective of cell morphology, as shown in Appendix A, the cells were observed to gradually shrink, becoming round and detached, which is consistent with the morphological characteristics of apoptosis.

### 2.5. Mitochondrial Membrane Potential (MMP) Analysis and ROS Analysis

Evidence has confirmed that **Na[Os^VI^(N)(tpm)_2_]** induces apoptosis via a caspase-3/9-dependent pathway. To further investigate the upstream signaling pathways in HepG2 cells, the cells were exposed to **Na[Os^VI^(N)(tpm)_2_]** to detect the MMP and levels of ROS production.

The red/green fluorescence of 5,5′,6,6′-tetrachloro-1,1′-3,3′-tetraethyl-benzimidazolylcarbocyanine iodide (JC-1), a mitochondria-selective fluorescent probe, was detected by confocal microscopy [53] (Figure 5A) and flow cytometry (Figure 5B). When the mitochondrial membrane potential is high, JC-1 accumulates in the matrix of the mitochondria to form polymers (J-aggregates) with red fluorescence. When the mitochondrial membrane potential is low, JC-1 exists as a monomer with a green fluorescence signal. As shown in Figure 5A, with increasing concentrations of **Na[Os^VI^(N)(tpm)_2_]**, the red fluorescence decreased and the green fluorescence increased. Significant red fluorescence was observed in HepG2 cells in the control group and only green fluorescence was detected in the positive control group treated with carbonyl cyanide m-chlorophenylhydrazone (CCCP). The decline in mitochondrial membrane potential (MMP) indicated that the early stages of apoptosis were induced by **Na[Os^VI^(N)(tpm)_2_]**.

Reactive oxygen species (ROS) are the main molecules produced by oxidative stress in the body and have been recognized as important factors in tumorigenesis, tumor development, and tumor recurrence. In recent years, studies have found that ROS and cell apoptosis are closely related. Therefore, progressive concentrations of **Na[Os^VI^(N)(tpm)_2_]** (3.25, 6.5, or 13 μM) were added to HepG2 cells for 3 h and an inverted fluorescence microscope was used to detect the oxidation of the sensitive fluorescence probe DCFH-DA by measuring the fluorescence intensity to test the content of intracellular ROS. As shown in Figure 6, compared with the negative control group, the fluorescence intensity of each treatment group obviously increased, which indicated that the complex promoted HepG2 cells to release ROS. This result is in accordance with proteomics analysis.

### 2.6. Anticancer Activity of **Na[Os^VI^(N)(tpm)_2_]** In Vivo

To reveal the anticancer activity of **Na[Os^VI^(N)(tpm)_2_]** in vivo, we examined the therapeutic effects of **Na[Os^VI^(N)(tpm)_2_]** in a human cancer xenografted in nude mouse model (Figure 7). We successfully established a xenograft tumor model in nude mice using HepG2 cells. HepG2 cell-bearing nude mice were randomly divided into the vehicle control group, two **Na[Os^VI^(N)(tpm)_2_]**-treated groups (0.2 and 1 mg/kg), and one cisplatin-treated group (1 mg/kg). Mice were treated with the different complexes via intravenous tail injection every 4 days. The tumor volumes and the mouse weights were measured every 2 days. After 24 days of treatment, the tumors from each group were excised and weighed. As shown in Figure 7A,B, the mice treated with **Na[Os^VI^(N)(tpm)_2_]** displayed a significant reduction in both tumor volume and weight compared with the vehicle control group. Dose-dependent inhibition of tumor growth after **Na[Os^VI^(N)(tpm)_2_]** treatment at 0.2 and 1 mg/kg showed relative tumor regression rates of 30.6% and 55.8%, respectively. The inhibitory effect of **Na[Os^VI^(N)(tpm)_2_]** on tumor growth was similar to that of cisplatin (57.8%) at the same concentration. Moreover, the body weights of the mice treated with **Na[Os^VI^(N)(tpm)_2_]** did not decrease significantly (Figure 7C), indicating that this complex possesses low toxicity compared with cisplatin. Together, our results suggest that **Na[Os^VI^(N)(tpm)_2_]** retains anticancer activity in vivo.

## 3. Materials and Methods

### 3.1. Instrumentation

UV/Visible spectra were recorded on a Shimadzu UV2450-2550 spectrophotometer (Shimadzu, Kyoto, Japan) in 1 cm cuvettes. The infrared spectrum was obtained from KBr plates using a Nicolet AVATAR 360 FTIR spectrophotometer (Nicolet, Madison, WI, USA). ^1^H-NMR spectrum was recorded on a Bruker DPX 400 spectrometer (Bruker, Karlsruhe, Germany). Elemental analysis was performed on a Vario EL cube CHNs analyzer (Elementar, Frankfort, Germany). X-ray crystallography was carried out with a SMART CCD (Bruker, Karlsruhe, Germany). ESI mass spectra were recorded on a PE-SCIEX API 365 triple quadruple mass spectrometer (AB Sciex, Boston, MA, USA).

### 3.2. Materials

In this study, 3-(4,5-Dimethyl-2-thiazolyl)-2,5-diphenyl-2H-tetrazolium bromide (MTT) and cisplatin were purchased from Alfa and used as received. FITC Annexin V Appoptosis Detection Kit I was purchased from BD Pharmingen™ (Lake Franklin, NJ, USA). Reactive oxygen species assay kit and mitochondrial membrane potential assay kit with JC-1 were obtained from Beyotime. β-Actin (CST, 13E5, #4970), Cyclin B1 (CST, D5C10, #12231), Caspase 3 (CST, D3R6Y, #14220), Cl-PARP1 (Abcam, ab4830), Cl-caspase 3 (Abcam, ab32042), Caspase 9 (Abcam, ab202068), and phospho-CDC2/CDK1 (R&D,Y15) were used as primary antibodies and prepared by 1:1000. HRP AffiniPure Goat Anti-Rabbit (BOSTER) was used as a secondary antibody. CST (Cell Signaling Technology, Danvers, MA, USA).

### 3.3. Synthesis of Complex

Here, [^n^Bu_4_N][Os^VI^(N)(Cl)_4_] (235.2 mg, 0.4 mmol) and [5-(Thien-2-yl)-1H-pyrazol-3-yl]methanol) (144.2 mg, 0.8 mmol) were fixed and 15 mL of methanol was added at room temperature, then it was stirred and, after they were completely dissolved, 5 mg/mL of NaOH of methanol solution was added to adjust the pH to 9. The mixture continuously stirred at room temperature for 12 h; the resulting yellow precipitate was filtered out using methanol washing and vacuum drying. The filtrate was volatilized at room temperature. After a few weeks, the needle yellow transparent crystal was obtained. Yield: 81.6%. IR(KBr, v/cm^−1^): ν (Os≡N) 1107 cm^−1^; ν (C=N) 146 cm^−1^; ν (C(=C)-H) 3125 cm^−1^. ^1^H NMR (400 MHz, *d*_6_-DMSO): δ 7.46 (s, 1H), δ 7.35(s, 1H), δ 7.07 (m, 1H), δ 5.2(d,1H), δ 4.9 (d, 1H). CHN, found: C, 33.04; H, 2.50; N, 11.41; S, 10.48. Calcd. for C_16_H_12_N_5_NaO_2_S_2_Os•CH_3_OH: C, 33.16; H, 2.62; N, 11.37; S, 10.42. ESI-MS: *m*/*z* = −562, [Os^VI^(N)(tpm)_2_]^−^.

### 3.4. X-ray Crystallography

Suitable single crystals were mounted with glue at the end of a glass fiber. X-ray diffraction data were collected on a XtaLab PRO MM007HF DW Diffractometer System equipped with a MicroMax-007DW Micro Focus X-ray generator and Pilatus 200 K silicon diarray detector (Rigaku, Tokyo, Japan, Cu K*α*, *λ* = 1.54184 Å) under 293 K. Data reductions were performed on CrysAlisPro 1.171.39.28b (Rigaku OD, 2015). Structure solution was carried out using SHELXT and refinement with SHELXL, within the OLEX2 graphical interface [54,55,56]. Restraints (SADI and DFIX) were applied for disordered methanol molecules. All non-hydrogen atoms were refined first isotropically and then anisotropically. All of the hydrogen atoms of the ligands were placed in calculated positions with fixed isotropic thermal parameters and included in the structure factor calculations in the final stage of full-matrix least-squares refinement. CCDC no. 2009053 contains the supplementary crystallographic data for the complex **Na[Os^VI^(N)(tpm)_2_]**•CH_3_OH. All data can be obtained free from The Cambridge Crystallographic Data Centre.

### 3.5. Stability of **Na[Os^VI^(N)(tpm)_2_]** in DMSO and in Aqueous Solutions in the Absence or Presence of GSH

The stability of **Na[Os^VI^(N)(tpm)_2_]** was examined by UV/Vis absorption spectra. The complex was dissolved in DMSO as stock solution. The UV/Vis absorption spectra were recorded at different time intervals. **Na[Os^VI^(N)(tpm)_2_]** was further diluted with PBS (3% DMSO) to give a final concentration of 30 µM and incubated at room temperature. The UV/Vis absorption spectra were recorded at different time intervals. **Na[Os^VI^(N)(tpm)_2_]** was incubated with GSH in PBS (3% DMSO) at room temperature. The UV/Vis absorption spectra were recorded at different time intervals.

### 3.6. Cell Culture Conditions

Cell lines used in this work including cervical epithelioid carcinoma (HeLa), liver hepatocellular carcinoma (HepG2) and its stem cell (HepG2-stem), lung carcinoma (A549) and its cisplatin-resistant daughter cell (A549CIS), ovarian carcinoma (A2780), and breast adenocarcinoma (MDA-MB-231) were kept in 10 cm^2^ culture plates at 37 °C/5% CO_2_. HeLa, A549, A549CIS, and MDA-MB-231 cells were cultured in Dulbecco’s modified Eagle’s medium (DMEM; Gibco) and HepG2 and A2780 cells were cultured in Roswell Park Memorial Institute (RPMI 1640; Gibco), supplemented with 1% penicillin-streptomycin (Gibco), 1% GlutaMAX (Gibco), and 10% fetal bovine serum (Gibco). Human normal liver cell line LO-2 (HL-7702) was obtained from the Cell Bank of the Chinese Academy of Sciences (Shanghai, China). LO-2 was cultured in RPMI 1640 and supplemented with 1% penicillin-streptomycin, 1% GlutaMAX, and 20% fetal bovine serum.

### 3.7. MTT Assay

Cells were rinsed with PBS and detached with 0.25% trypsin-EDTA (Gibco). Then, cells with a density of 5 × 10^4^ cells/mL were counted and added into the 96-well plate. Cells were incubated for 24 h and then the solution containing the tested compound and positive control cisplatin was added. The compound was solved in DMSO as stock solution and then diluted with culture medium, and the percentage of DMSO was under 1%. Cisplatin was solved in 0.9%NaCl and then diluted with culture medium. After 48 h, drug solutions were replaced by 0.5 mg/mL MTT solution. After incubation for 1.5 h at 37 °C, the old solution was removed and DMSO was added, then the 96-well plate was vortexed for 15 min in the dark. OD values at a wavelength of 570 nm were read by Infinite M200 (Swiss, Tecan). The curve was fitted using the logarithmic interpolation in Origin software and the half inhibition concentration (IC_50_) was calculated. Each experiment was repeated three times.

### 3.8. Proteomic Analysis

HepG2 cells were treated with **Na[Os^VI^(N)(tpm)_2_]** at a concentration of 6.5 μM for 8 and 24 h in a CO_2_ incubator at 37 °C. Equal amounts of DMSO were added to HepG2 cells as a positive control. The cells were then rinsed with cold PBS three times. Lyse cells with freshly prepared 8 M urea lysis buffer (8 M urea in 20 mM Tris-HCl, pH 8.0) contained protease inhibitor, 1 mM PMSF (phenylmethanesulfonylfluoride), 1 mM Na_3_VO_4_, and 1 mM NaF. They were kept on ice for 10 min and scraped to collect cell lysis into a 1.5 mL tube. The sample was clarified by centrifugation at 13,000–13,500 rpm for 15 min at 4 °C. The clarified supernatant may be stored at −80 °C and protein concentration was then measured. The volume of samples containing a certain amount of proteins (50 μg) was measured. Fourfold volume of ice-cold acetone was added to the sample solution (acetone is kept at −20 °C). The sample solution was immediately well mixed with acetone and keep at −20 °C for 30 min to overnight. Centrifugation with 13,000 rpm at 4 °C for 20 min was performed. The protein pellet was finally dried and stored at −80 °C. The protein pellet was suspended in urea buffer and then denatured protein at 60 °C for 10 min. DTT was added to a final concentration of 5 mM. It was incubated at room temperature for 20 min. Iodoacetamide was added to a final concentration of 25 mM at room temperature for 30 min in the dark. Then, 100 mM of Tris-HCl (pH 8.0) was added to the sample to dilute urea to 1 M. Trypsin was added to the sample at a ratio of 1:50–1:100 (trypsin/protein), following incubation at 37 °C for 16 h. Proteolysis was stopped by adding formic acid to a final concentration of 5% and centrifuging at 14,000 rpm for 15 min. The resulting peptides were desalted through StageTips [57,58]. For each sample, three biological replicates were prepared. The samples were re-dissolved with H_2_O (containing 0.1% formic acid, *v*/*v*) for subsequent HPLC-MS/MS analysis.

MS analysis was performed with a LTQ Orbitrap Velos Orbitrap MS (Thermo) connected online with an HPLC. The analytical column was a self-packed PicoTip^®^ column (360 μm outer diameter, 75 μm inner diameter, 15 μm tip, New Objective) packed with 10 cm length of C18 material (ODS-A C18 5-μm beads, YMC) with a high-pressure injection pump (Next Advance). The mobile phases of HPLC are A (0.1% formic acid in HPLC grade H_2_O, volume percentage) and B (0.1% formic acid in HPLC grade acetonitrile, volume percentage). Three micrograms of the sample were loaded onto the analytical column by the auto-sampler and rinsed with 2% B for 6 min and, subsequently, eluted with a linear gradient B from 2% to 40% for 120 min. For the MS analysis, LTQ-Orbitrap Velos mass spectrometer (Thermo Fisher Scientific) was operated in a data-dependent mode, cycling through a high-resolution (6000 at 400 *m*/*z*) full scan MS1 (300–2000 *m*/*z*) in Orbitrap followed by CID MS2 scans in LTQ on the 20 most abundant ions from the immediate preceding full scan. The selected ions were isolated with a 2 Da mass window and put into an exclusion list for 60 s after they were first selected for CID.

The differentially expressed proteins (1.5-fold changes after the **Na[Os^VI^(N)(tpm)_2_]** treatment) identified by mass spectrometry analysis were subjected to Gene Ontology (GO) analysis, as described previously [59]. The top six GO terms under the biological process category were retrieved for profiling the functional enrichment of up- and downregulated gene products, respectively.

### 3.9. Cell Cycle Analysis

HepG2 cells were rinsed with PBS and detached with 0.25% trypsin-EDTA (Gibco). Then, cells with a density of 1 × 10^5^ cells/mL were counted and 2 mL was added into the six-well plate. After being cultured for 24 h, **Na[Os^VI^(N)(tpm)_2_]** at the indicated concentrations was added and reacted for 12 h. Cells were collected and 70% pre-cooled ethanol was used to fix cell at 4 °C for 24 h. Before staining, cells were washed twice with PBS. After being re-suspended with PBS, RNase and propidium iodide were added and stained for 30 min in the dark, and cells were prepared for analysis. The cells were then determined by a flow cytometer (BD Accuri™ C6, Franklin Lakes, NJ, USA).

### 3.10. Apoptosis Evaluation

HepG2 cells were rinsed with PBS and suspended with 0.25% trypsin-EDTA (Gibco). Then, cells with a density of 1 × 10^5^ cells/mL were counted and 2 mL was added into the six-well plate. After being cultured for 24 h, **Na[Os^VI^(N)(tpm)_2_]** at the indicated concentrations was added and reacted for 24 h. HepG2 cells were suspended with EDTA-free trypsin (Gibco) and rinsed twice with cold PBS. Then, cells were resuspended in 1X binding buffer. Then, 5 μL FITC Annexin V and 5 μL PI were added to 100 μL of the solution (1 × 10^5^ cells). The cells were gently vortexed and stained for 15 min at RT in the dark. Then, 400 μL of 1X binding buffer was added to each sample. It was analysed by flow cytometry for 1 h. Flow cytometric analysis was performed with a flow cytometer (BD Accuri™ C6, USA).

### 3.11. Western Blot Analysis

HepG2 with a density of 1 × 10^6^ cells/mL was counted and 2 mL/well was added into the six-well plate. After being cultured for 24 h, **Na[Os^VI^(N)(tpm)_2_]** at the indicated concentrations was added and reacted for 24 h. Cells were scraped with a scraper, collected, and washed twice with cold PBS. Cells were lysed with a cell lysis buffer, of which the main active component is 1% Triton X-100, with protease inhibitor phenylmethanesulfonyl fluoride (PMSF). Protein quantifications were measured using the BCA Protein Assay Kit (Beyotime) by Infinite M200 (Männedorf, Swiss, Tecan). Equal amounts of cellular proteins were mixed with SDS-PAGE Sample Loading Buffer, 5X (Beyotime), boiled at 95 °C for 5 min, and run on 12–15% separation gel. Protein was transferred to the nitrocellulose membrane (BOSTER). The membrane was blocked with 5% nonfat milk in TBST (1X, 0.1% Tween-20) for 40 min at RT, washed with TBST for 5 s, and then probed with primary antibody at 4 °C overnight followed by secondary antibody. Next, the ECL Plus detection kit (Beyotime) was added and the membrane was visualized using High ChemiDoc XRS (Bio-Rad ChemiDoc XRS+, Hercules, CA, USA).

### 3.12. Analysis of Mitochondrial Membrane Potential (MMP)

#### 3.12.1. Flow Cytometry

In this study, 2 mL HepG2 cells with a density of 2 × 10^5^ cells/mL were counted and added into the six-well plate. After being cultured for 24 h, **Na[Os^VI^(N)(tpm)_2_]** at the indicated concentrations was added and the cells were further cultured for 12 h. The cells were collected and resuspended in 1 mL JC-1 dyestuff and placed into a cell incubator for 20 min. Subsequently, the cells were rinsed twice with pre-cooling JC-1 staining buffer and measured by flow cytometry; 10,000 cells were acquired for each sample.

#### 3.12.2. Confocal Microscopy

HepG2 cells were seeded in a glass bottom dish (35 mm dish with a 10 mm bottom well, 4 × 10^5^ cells/well). After being cultured for 24 h, **Na[Os^VI^(N)(tpm)_2_]** at the indicated concentrations was added and the cells were further cultured for 12 h. After removal of culture medium, cells were washed with PBS. Then, 1 mL fresh medium and 1 mL JC-1 dyestuff were added to each dish and placed into a cell incubator for 20 min. Cells were washed twice with pre-cooling JC-1 staining buffer and incubated with 2 mL fresh medium. LSM 880 confocal laser scanning microscope was adopted for observation and photo taking.

### 3.13. Detection of Intracellular Reactive Oxygen Species (ROS) by DCFH-DA

HepG2 cells were rinsed with PBS and suspended with 0.25% trypsin-EDTA. Then, cells with a density of 1 × 10^5^ cells/mL were counted and 2 mL was added into the six-well plate. After being cultured for 24 h, different concentrations of drugs were added. Then, the cells were placed in an incubator for further 3 h. The old liquid was replaced with the diluted DCFH-DA probe (10 μmol/L). The volume is 1 mL per well. After 20 min, the cells were rinsed three times with serum-free cell culture medium. The green fluorescence of each sample was then observed by an inverted fluorescence microscope.

### 3.14. Cell Morphology Observed by an Inverted Microscope

HepG2 cells were rinsed with PBS and suspended with 0.25% trypsin-EDTA (Gibco). Then, cells with a density of 1 × 10^5^ cells/mL were counted and 2 mL was added into the six-well plate. After being cultured for 24 h, different concentrations of **Na[Os^VI^(N)(tpm)_2_]** (6.5, 13, and 26 μmol/L, respectively) were added. Then, the cells were placed into an incubator for a further 3 h and the cell morphology of each group was directly observed with an inverted microscope.

### 3.15. Xenograft in Nude Mice

Nude mice were obtained from the Hunan SJA Laboratory Animal Co., Ltd. (Changsha, China). HepG2 cells were suspended with 0.1 mL serum free culture medium and were injected subcutaneously in the right blank of nude mice (4 weeks old). Tumour size was measured with digital caliper and calculated as the following formula: tumour volume (mm^3^) = longest diameter (mm) × [shortest diameter (mm)]^2^/2. After 5 days, HepG2-bearing nude mice were randomly assigned into four different treatment groups. Mice were injected with complex (solvent: PBS with 1% DMSO) at the dose of 0.2 and 1 mg/kg body weight and 1 mg/kg cisplatin (solvent: PBS) via tail intravenous injection every 4 days. The tumour volume and the mice weight were detected every 2 days. The treated mice were scarified at the end of the studied period (29 days) with anatomical separation of tumour nodules. The tumours were weighed and photographed.

The inhibition rate of tumour growth was determined using the following formulas:

Tumour growth inhibition rate (%) = (1 − mean tumour weight of the treatment/mean tumour weight of the negative group) × 100%.

All measurements were expressed as mean ± SD, using SPSS 10.0 for statistical analysis.

## 4. Conclusions

In summary, an osmium(VI) nitrido complex **Na[Os^VI^(N)(tpm)_2_]** bearing a pyrazolate ligand was synthesized and characterized. It exhibits cytotoxicity against cancer cell lines, cancer stem cells, and cisplatin-resistant cells. The compound induces apoptosis and cell cycle arrest towards HepG2 cells. In addition, caspase activation and oxidative stress in the cells after **Na[Os^VI^(N)(tpm)_2_]** treatment suggest that apoptosis occurs through intrinsic (mitochondrial) pathways. It can regulate protein homeostasis by mobilizing different functional branches of the protein network at different time points during treatment. Protein-transportation-associated proteins were significantly downregulated at earlier time points after **Na[Os^VI^(N)(tpm)_2_]** treatment. Furthermore, DNA-metabolism-associated and oxidative-stress-associated proteins were significantly downregulated and upregulated, respectively, at later time points after **Na[Os^VI^(N)(tpm)_2_]** treatment, suggesting that the complex can decrease DNA repair capacity while increasing DNA damage accumulation and DNA replication pressure, which may increase oxidative stress in the cells. Our study reveals a new high-valent osmium complex as a promising anticancer agent candidate targeting protein homeostasis.

## Figures and Tables

**Figure 1 ijms-23-12779-f001:**
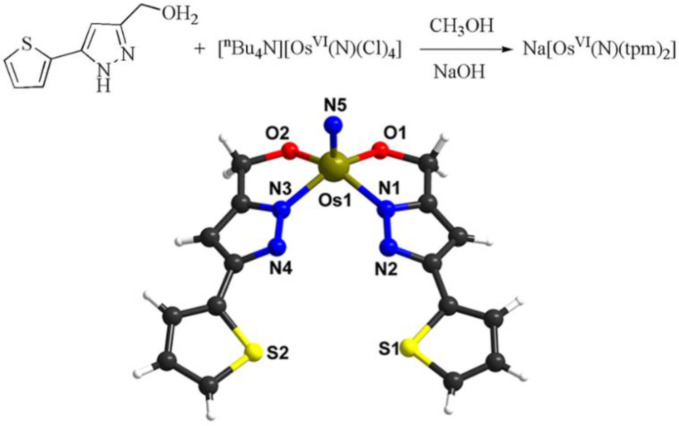
Synthesis and X-ray crystal structure of **Na[Os^VI^(N)(tpm)_2_]**.

**Figure 2 ijms-23-12779-f002:**
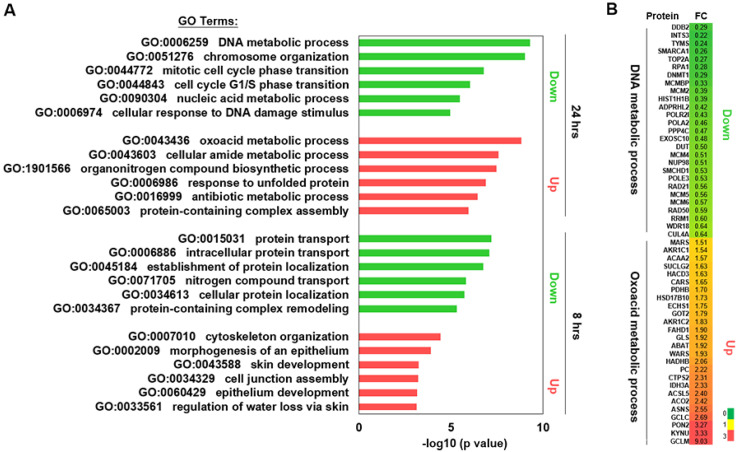
(**A**) Gene ontology analysis for downregulated and upregulated proteins after 8 or 24 h of treatment with **Na[Os^VI^(N)(tpm)_2_]**. (**B**) The heat map shows the fold change in expression of the downregulated proteins in terms of the DNA metabolic process and the upregulated proteins in terms of the oxoacid metabolic process after 24 h **Na[Os^VI^(N)(tpm)_2_]** treatment from panel (**A**).

**Figure 3 ijms-23-12779-f003:**
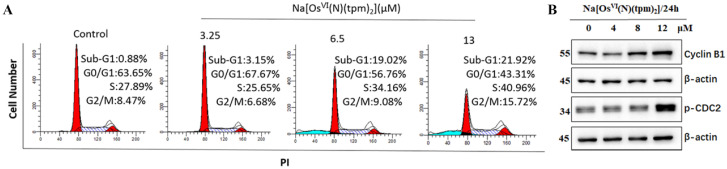
(**A**) Cell cycle distribution of HepG2 cancer cells exposed to the **Na[Os^VI^(N)(tpm)_2_]** complex for 12 h. (**B**) The expression levels of cyclin B1 and p-CDC2 in HepG2 cells were assessed by Western blot after treatment with **Na[Os^VI^(N)(tpm)_2_]** for 24 h.

**Figure 4 ijms-23-12779-f004:**
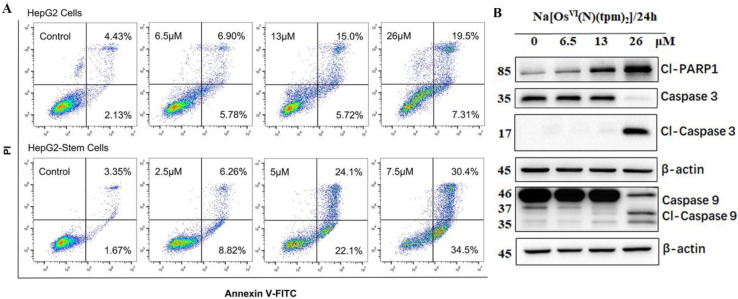
(**A**) Apoptosis analysis of HepG2 cancer cells and HepG2-stem cancer stem cells after 24 h of exposure to the **Na[Os^VI^(N)(tpm)_2_]** complex as determined by flow cytometry using Annexin V-FITC/PI staining. (**B**) The expression levels of several proteins involved in apoptosis in HepG2 cells were assessed by Western blot after treatment with **Na[Os^VI^(N)(tpm)_2_]** for 24 h.

**Figure 5 ijms-23-12779-f005:**
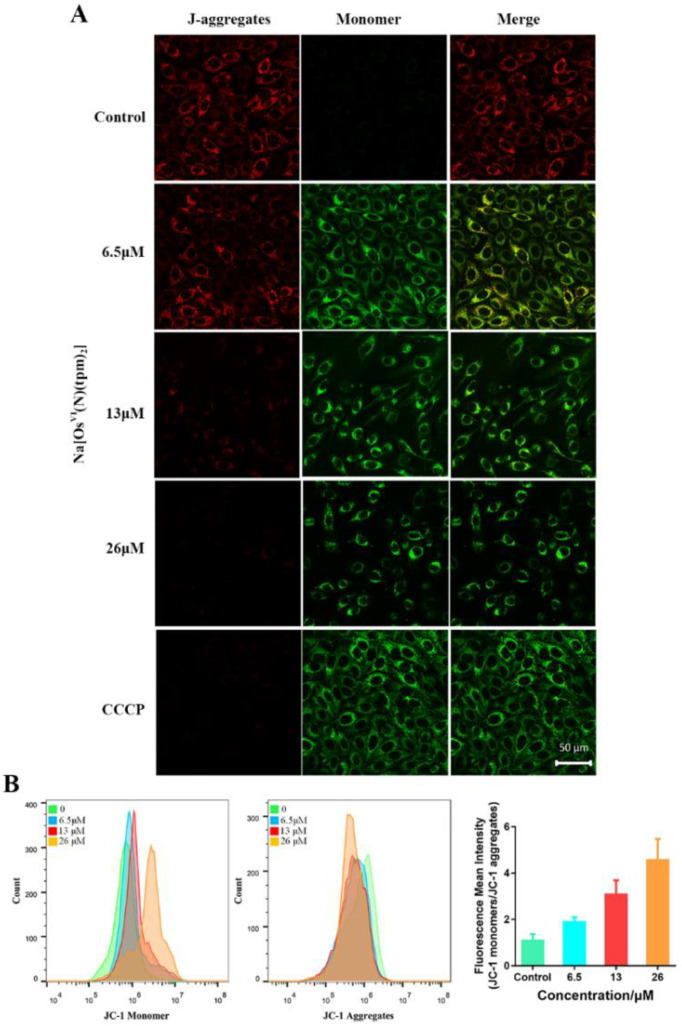
(**A**) Effect of **Na[Os^VI^(N)(tpm)_2_]** (6.5, 13, or 26 µM) incubated for 12 h, a blank control, and the positive control carbonyl cyanide m-chlorophenylhydrazone (CCCP) (10 µM) incubated for 20 min on the mitochondrial membrane potential of HepG2 cells by confocal microscopy. (**B**) Effect of the complex **Na[Os^VI^(N)(tpm)_2_]** (6.5, 13, or 26 µM) incubated for 12 h on the mitochondrial membrane potential of HepG2 cells by flow cytometry and quantitative analysis with histograms.

**Figure 6 ijms-23-12779-f006:**
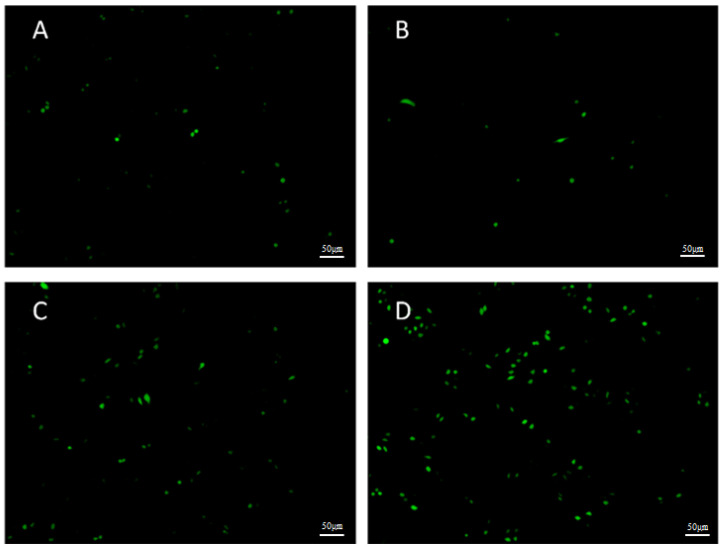
Fluorescent microscopic photographs of DCFH-DA staining in HepG2 cells treated with **Na[Os^VI^(N)(tpm)_2_]** for 3 h. (**A**) Control; (**B**) 3.25 μM; (**C**) 6.5 μM; and (**D**) 13 μM.

**Figure 7 ijms-23-12779-f007:**
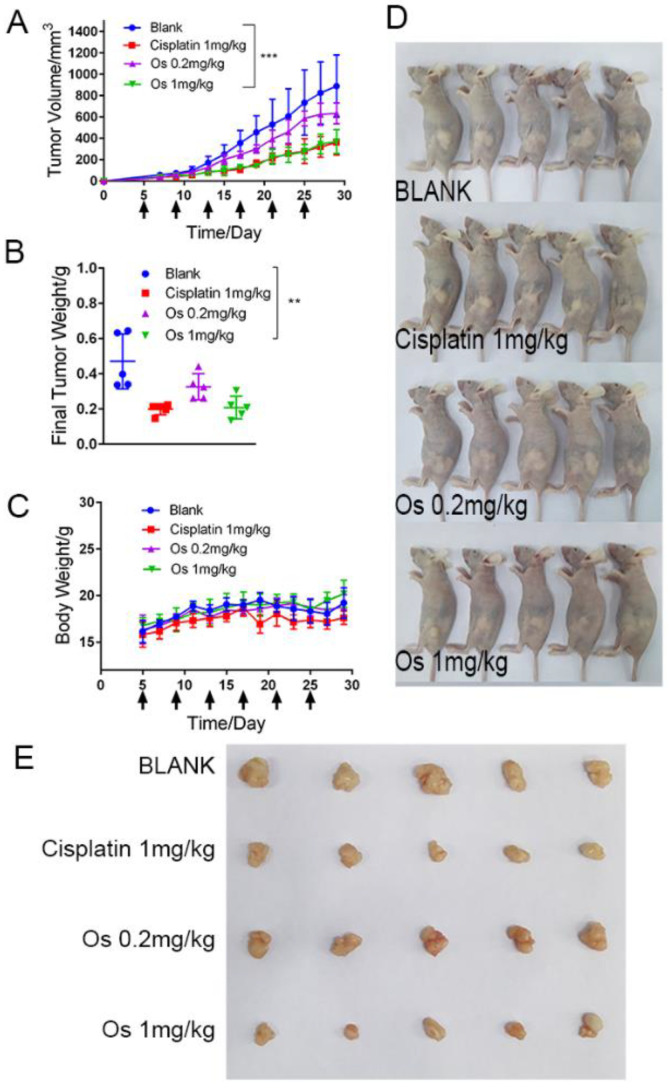
In vivo anticancer activity in nude mice bearing HepG2 tumor xenografts. (**A**) Tumor growth curves. Tumor volume of the 1 mg/kg **Na[Os^VI^(N)(tpm)_2_]** treated group vs. vehicle control group, *** *p* < 0.001. (**B**) Final tumor weights. Tumor weight of the 1 mg/kg **Na[Os^VI^(N)(tpm)_2_]** treated group vs. vehicle control group, ** *p* < 0.01. (**C**) Animal body weight growth curves. (**D**) Photograph of the mice in all groups on day 29. (**E**) Photograph of the tumors in all groups on day 29. **Os** represents **Na[Os^VI^(N)(tpm)_2_]**. Data are presented as the mean ± S.D. (*n* = 5).

**Table 1 ijms-23-12779-t001:** The half inhibition concentration (IC_50_) of **Na[Os^VI^(N)(tpm)_2_]** and cisplatin on human cell lines for 48 h.

Cell Lines	Na[Os^Ⅵ^(N)(tpm)_2_]	Cisplatin
HeLa	10.8 ± 1.0	13.4 ± 0.7
A2780	5.6 ± 0.4	15.9 ± 0.9
MDAMB231	9.1 ± 0.6	11.3 ± 1.6
HepG2	6.5 ± 1.1	4.9 ± 0.3
HepG2-stem	8.2 ± 0.8	3.3 ± 0.2
LO-2	28.5 ± 5.1	5.0 ± 0.3
A549	11.5 ± 0.8	12.3 ± 1.5
A549CIS	15.3 ± 2.0	149.6 ± 10.6
RI ^1^	1.3	12.2

^1^ RF: resistance factor calculated as RI = IC_50_(A549CIS)/IC_50_(A549).

## Data Availability

The MS data have been deposited in the ProteomeXchange Consortium via the iProX repository with the link https://www.iprox.cn/page/PSV023.html;?url=1662181205956ARVj. The password is 0xDU (accessed on 28 August 2022).

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
