# Peer review of "A Pyrazolate Osmium(VI) Nitride Exhibits Anticancer Activity through Modulating Protein Homeostasis in HepG2 Cells"

_ijms, 2022, doi:10.3390/ijms232112779_

Round 1

Reviewer 1 Report

The manuscript entitled „A new pyrazolate osmium(VI) nitrido complex exhibits anticancer activity through modulating protein homeostasis” ijms-1967005 submitted to the International Journal of Molecular Sciences presents interesting and significant results in the context of cancer treatment because they indicated the new type of compound with significant anticancer potential which was even slightly better than this of conventional cytostatic, cisplatin. It is important that the therapeutic potential of tested compound was indicated both in in vitro and in vivo studies. The methods were reliable and mostly described carefully, with some exceptions indicated below. Hence, taking into account this and other minor failures and shortcomings indicated below, I recommend minor revision.

  • I suggest indicating more precisely in the title of manuscript towards which type of cancer tested complex showed the best activities.
  • If it is possible concerning the acceptable number of words in abstract, I suggest introducing short comparison of the results on anticancer activity of cisplatin and tested complex obtained in this study.
  • In my opinion periods at the end of a sentence should be place after references numbering.
  • The Authors indicated that the density of cell cultures used in study on anticancer ability of tested complex was 5x104cell/ml. In my opinion, this cell culture density is used rather in proliferation assay than in viability assays. Hence I suggest changing the statements like “viability”, “cytotoxicity” etc. to “proliferation” or “antiproliferative”, etc.
  • I suggest indicating in table caption that the presented values are IC50.
  • I suggest placing the results for A549 and A549Cis, and RI one by one at the end of table as corresponding each other.   
  • I suggest introducing the explanations of all used abbreviations, both in the main text (e.g. NTS3; hSSB1; SMARCA1; DDB; etc) and figure captions (e.g. Cl casapase).
  • I suggest changing the expression of complex concentration in 0.5, 1.0, 2.0 times of IC50 to concentrations in “µM” which is more informative.
  • In my opinion it would be better if there are two separate figures corresponding with apoptosis and cell cycle analysis. I suggest combining appropriate flow cytometry histograms/dot plots with blots. Unfortunately, I couldn’t see cell cycle histograms which were indicated as presented as supplementary data.    
  • I suggest making quantitative analysis of MMP flow cytometry assay and combined the obtained results along with histograms with microscopic analysis which is mainly considered as qualitative analysis.
  • I suggest adding “…and ROS production analysis” in title of 2.5 paragraph.
  • The results presented in photos indicated the ROS production (Figure 6) but this result does not allow clearly to state that ROS are responsible for apoptosis induction (line 265). Of course, this could not be excluded but it would require additional assays for example with ROS scavengers.
  • I suggest avoiding the statement that cells were “digested with 0.25% trypsin-EDTA (e.g. line 417)”. The phrase “digested” suggests destroyed, broken. Cells were rather “detached” from the culture dishes without any signs of destruction which would be a result of digestion.   
  • Please indicate the manufactures of antibodies used in Western blot assay, and their detail description e.g. product number, the used concentration, etc.  
  • The conclusion that “…The complex exhibits anticancer activity
    against cancer cell lines, cancer stem cells, and cisplatin-resistant cells by inducing cell apoptosis and cell cycle arrest…., (500-502) etc” is too general and exaggerated, not established in this study because apoptosis and cell cycle were not examined in cancer stem cells and cisplatin-resistant cells but only in one cancer cell line, i.e. HepG2.
  • In my opinion, there are no evidence in this manuscript which would suggest and indicate that apoptosis in HepG2 cells was induced by tested complex via extrinsic, receptor-dependent pathway (conclusion, line 503).   

Reviewer 2 Report

The manuscript is interesting and well structured. The synthesized compound has been extensively studied from different points of view, mainly biological. It would be interesting to add some details also on the chemical reactivity of this compound, since it belongs to a class of unusual and interesting compounds.

 Some issues to be solved:

1.                   The synthesis should be described in a little more detail

2.                   The ESI-MS spectrum of the compound should be reported together with the appropriate attributions. The isotopic pattern of osmium is complex, and a description of the mass spectrum is desirable

3.                   Some mass spectrum collected during the metabolomic study should be reported because of sure interest

4.                   In the caption of the figures of the absorption spectra some experimental details are missing (molar concentration, temperature, optical path,...)

It is advisable that the manuscript is read by a native English speaker, sometimes there are typos
